# How to Prevent Catheter-Associated Urinary Tract Infections: A Reappraisal of Vico’s Theory—Is History Repeating Itself?

**DOI:** 10.3390/jcm11123415

**Published:** 2022-06-14

**Authors:** Stefania Musco, Alessandro Giammò, Francesco Savoca, Luca Gemma, Paolo Geretto, Marco Soligo, Emilio Sacco, Giulio Del Popolo, Vincenzo Li Marzi

**Affiliations:** 1Unit of Neuro-Urology, Azienda Ospedaliera Careggi, 50134 Florence, Italy; muscos@aou-careggi.toscana.it (S.M.); delpopolog@aou-careggi.toscana.it (G.D.P.); 2Unit of Neuro-Urology, Orthopaedic Trauma Center (CTO)-Spinal Unit Hospital, Città Della Salute e Della Scienza, 10126 Turin, Italy; giammo.alessandro@gmail.com (A.G.); paolo.gere@gmail.com (P.G.); 3Unit of Urology, Cannizzaro Hospital, 95122 Catania, Italy; francescosavoca@virgilio.it; 4Unit of Urological Robotic Surgery and Renal Transplantation, Azienda Ospedaliera Careggi, 50134 Florence, Italy; gemmaluca.dr@gmail.com; 5Department of Experimental and Clinical Medicine, University of Florence, 50121 Florence, Italy; 6Unit of Obstetrics and Gynecology, Ospedale Maggiore di Lodi, 26900 Lodi, Italy; dr@marcosoligo.it; 7Unit of Urology, Fondazione Policlinico Universitario Agostino Gemelli IRCCS, Università Cattolica del Sacro Cuore, 00168 Rome, Italy; emilio.sacco@unicatt.it

**Keywords:** CAUTI, indwelling urinary catheter, neurogenic bladder, geriatric incontinence, endovesical instillation, long-term care

## Abstract

New, contextualized modern solutions must be found to solve the dilemma of catheter-associated urinary infection (CAUTI) in long-term care settings. In this paper, we describe the etiology, risk factors, and complications of CAUTI, explore different preventive strategies proposed in literature from the past to the present, and offer new insights on therapeutic opportunities. A care bundle to prevent CAUTI mainly consists of multiple interventions to improve clinical indications, identifying a timeline for catheter removal, or whether any alternatives may be offered in elderly and frail patients suffering from chronic urinary retention and/or untreatable urinary incontinence. Among the various approaches used to prevent CAUTI, specific urinary catheter coatings according to their antifouling and/or biocidal properties have been widely investigated. Nonetheless, an ideal catheter offering holistic antimicrobial effectiveness is still far from being available. After pioneering research in favor of bladder irrigations or endovesical instillations was initially published more than 50 years ago, only recently has it been made clear that evidence supporting their use to treat symptomatic CAUTI and prevent complications is needed.

## 1. Introduction

According to the National Healthcare Safety Network of Centers for disease control and prevention, catheter-associated urinary tract infections (CAUTIs) represent one of the major causes of infection associated with healthcare needs worldwide [1,2,3,4,5,6]. With the abuse and overuse of antibiotics, an invisible army of super-resistant bacteria has evolved, and many scientists agree that this may lead to a global catastrophe. As Giambattista Vico would have said, we are moving towards a “*new barbaric era”*—the post-antibiotic age. Nonetheless, Vico’s notion must be interpreted as a starting point to reflect on and recall from our past any elements, which may be helpful in finding contextualized modern solutions. The objective of this review is to (1) describe the etiology, risk factors, and complications of CAUTI, particularly those related to indwelling catheters for the management of urinary incontinence in elderly and neurogenic bladder; (2) explore the existing studies regarding preventive and prophylactic strategies; (3) focus on published experiences with endovesical instillation and/or bladder irrigation and possibly offer new insights on therapeutic opportunities.

## 2. Epidemiology and Definition

Urinary tract infections (UTIs) are the fourth most common type of healthcare-associated infection, with an estimated 93,300 UTIs in acute care hospitals in 2011, accounting for more than 12% of their reported infections [1]. Complications secondary to catheter-associated UTI (CAUTI) cause prolonged hospitalization, increased costs, and mortality with more than 13,000 deaths estimated per year in the United States [2].

Approximately 12–16% of adult hospital inpatients will have an indwelling urinary catheter at some time during their hospitalization, and each day the indwelling urinary catheter remains, a patient has a 3% to 7% increased risk of acquiring a CAUTI [3,4]. CAUTI accounts for over 1 million cases in the United States alone and almost 80% of the nosocomial infections worldwide [5,6]. Abulhasan et al. conducted a six-year prospective analysis of neurological ICU patients and found that they had documented CAUTI at a rate of 3 to 5.3 infections per 1000 urinary catheter days [7]. Every CAUTI episode has been estimated to cost approximately USD 600, contributing to nearly 131 million dollars in annual nationwide costs [8].

Most cases of bacteriuria in the context of an indwelling catheter are asymptomatic. However, international scientific societies are still seeking clarification on how and when to treat asymptomatic or paucisymptomatic UTIs in patients with an indwelling catheter [9].

The Infectious Disease Society of America (IDSA) has defined CAUTI through the following criteria: (1) indwelling urinary catheter for more than 48 h after insertion; (2) one sign or symptom including fever, suprapubic tenderness, costovertebral angle tenderness, urgency or dysuria; and (3) urine culture with more than 10^5^ colony-forming units (CFU)/mL of one bacterial species [10,11]. In addition to the aforementioned symptoms, patients with symptomatic UTI may generally present with chills, flank pain, altered mental status (if older than 65), hypotension, and evidence of systemic inflammatory response syndrome (SIRS) [10].

In 2015, the Association for Infection Control and Epidemiology (APIC) analyzed the 2009 criteria of the National Healthcare Safety Network (NHSN), seeking to respond to the gaps emerging from the experts’ opinions. The following issues were included: (1) yeast infections; (2) urine cultures with low concentrations (i.e., <100,000 colony-forming units CFU/mL); (3) signs of infection/inflammation on chemical–physical examination of the urine (e.g., leukocyturia) (4) fever (should not be considered the only criterion); (5) specific criteria based on population types (i.e., neurological, elderly).

Loeb et al. introduced an algorithm to reduce the prescription rate of antibiotics in long-term care for elderly and/or fragile patients [12]. According to the Loeb criteria, antimicrobial therapy should be considered only if one of the following is present: fever, new costovertebral angle pain, new onset of delirium, or stiffness.

Due to reduced sensitivity in neurological patients, a diagnosis could be even more difficult in these patient populations. The IDSA guidelines for CAUTI recognize this, and in their decision-making algorithms, they include specific symptoms of spinal cord injury (SCI) such as increased spasticity and the appearance of signs of autonomic dysreflexia (e.g., hypertension, piloerection, headache, flushing), but there is still limited evidence today on the sensitivity/specificity of these symptoms. Cloudy or foul-smelling urine is also often considered a sign of CAUTI, but no studies have outlined the clinical significance of these findings, even if they are of recent onset [11].

Studies on objective measures such as biomarkers of infection have been pursued to aid in diagnosis. A systematic review demonstrated that procalcitonin appears to be a promising marker for diagnosing and initiating treatment of CAUTIs and pyelonephritis [13]. The role of procalcitonin in adults is less evident, where instead interleukin-6 would seem to be more effective in distinguishing pyelonephritis in CAUTIs. However, no biomarker has so far proved useful in the diagnosis of CAUTIs in geriatric and neurological populations. Thus, subanalyses between the various population subtypes require further confirmation and scientific studies.

## 3. Etiology and Risk Factors

Several pathogens can be associated with urinary tract infections, of which Gram-negative bacteria are the most frequent. Uropathogenic *Escherichia coli* (UPEC) is the most common and related to more than 20% of CAUTIs [14]. Different physiopathological mechanisms have been described. Pili, flagella, and adhesins, which are expressed by UPEC and *Klebsiella pneumoniae*, allow bacterial adhesion and invasion of the urothelium. Type 1 pili, frequently expressed by UPEC, mediate the adhesiveness to mannosylated host cells [15].

Although Gram-positive bacteria are less frequently associated with UTIs, recent studies underline that *Staphylococcus*
*aureus* is often isolated in complicated infections, and catheterization is one principal risk factor for UTIs caused by *S. aureus*. Bladder catheterization causes chronic traumatism on urothelium and inflammation, with the subsequent release of host proteins such as fibrinogen, resulting in persistent colonization and infection [16]. A catheter’s internal and external surface can act as a scaffold for bacterial adhesion, proliferation, and biofilm creation [17]. Yeasts are frequently found in complicated UTIs: *Candida* spp. is responsible for more than 17% of CAUTI [13]. This percentage can be much higher in ICU-acquired symptomatic UTIs, where *Candida* spp. is found in more than 46% of cases [18]. Though the patient setting is an important risk factor for the onset of CAUTI, the dominant risk factors for CAUTI are the duration of catheterization, gender (>female), age >50 years or <17 years, diabetes or other comorbidities, renal impairment, and non-surgical disease [19,20]. Additionally, the non-adherence to catheterization sterility protocol and catheterization in a non-sterile environment, along with insufficient professional training of the inserter, have been associated with a higher risk of CAUTI [19,20].

An indwelling urinary catheter should be used only in selected cases and removed as soon as possible. In particular, the IDSA recommends its use only in the following cases: (a) urinary retention non-responders to conservative and/or invasive treatments or where it is not possible to use alternative methods of urinary drainage (intermittent catheterization or external catheters, urine containment devices); (b) urinary incontinence in patients with a terminal disease; (c) frequent and urgent need to monitor diuresis in critically ill subjects [4].

## 4. Types and Lifecycle of Indwelling Catheters

An indwelling catheter resides in the bladder for short or long periods of time and is generally inserted through the urethra. The risk of CAUTI increases over time mostly because of cross-contamination from the drainage bag and the rich microbial flora in the skin. Among the various approaches used to prevent CAUTI, specific urinary catheter coatings according to their antifouling and/or biocidal properties have been widely investigated. Antifouling coatings do not kill the microbes directly but prevent biofilm formation by means of steric or electrostatic repulsion and low surface energy [21]. Biocidal materials such as silver ions, triclosan, chlorhexidine, chlorine, tributyltin, nitric oxide, and antibiotics are designed to kill the microbes, protecting them from infection and encrustation development (Figure 1), but their effectiveness is still limited [22]. Antibiotic coatings are introduced as better alternatives to silver-alloy catheters because of their cytotoxicity. Nonetheless, they are burdened by bacterial-resistance issues [23]. Ideally, a catheter would offer holistic antimicrobial effectiveness and good safety, but further research is required to identify the optimal antimicrobial coating.

According to Meddings et al., to prioritize potential interventions and prevent CAUTI, the “lifecycle” of an indwelling catheter must be examined [24]. A catheter’s “lifecycle” (1) begins with its initial placement, (2) continues while it is in place, (3) ceases when removed, and (4) may start over when a new catheter is inserted. Two of the most important interventions that target unnecessary urinary catheter use are decreasing unnecessary placement and removing the urinary catheter as soon as possible (Figure 2) [25].

Avoiding unnecessary initial catheter placement is vital. Knoll et al., drew attention to this factor when placing the catheter and to possible alternatives such as external catheters or bedside urinals, using ultrasound to evaluate bladder emptying, and finding any further different approaches to finally disrupt the lifecycle [26].

The most important task after placing the catheter is maintaining awareness of its existence. A variety of interventions, including an electronic reminder, a daily checklist, and dedicated catheter nurses daily assessing the patient’s conditions are crucial to guarantee a shorter catheterization period [26].

When lifecycle stage 3 cannot be interrupted, the urinary catheter must be periodically replaced. However, there is insufficient evidence to support periodical catheter replacement besides the type of material rather than on demand (e.g., because of the onset of infection and/or obstruction) [27].

## 5. Complications of CAUTIs

CAUTIs are related to high economic burden and morbidity. When untreated, these infections can determine higher urinary tract infections with pyelitis and pyelonephritis and can lead to urosepsis and death. It was estimated that the mortality rate associated with CAUTIs is approximately 10% [28]. Moreover, bladder management with an indwelling urinary catheter can be complicated by several conditions, such as urethral damages (iatrogenic hypospadias, urethral fistulas) and the formation of bladder stones [29]. Almost 46–53% of patients managed with long-term indwelling urinary catheters can develop bladder stones and require surgical removal. Furthermore, the incidence of upper urinary tract calculi has been found to be higher than in the general population.

Calculi formation can be explained by the frequent colonization by *Proteus mirabilis,* a Gram-negative bacterium that is often implied in CAUTI, producing a dense biofilm and enzyme urease. Urease mediates the hydrolysis of the urinary urea into CO_2_ and NH_3_, raising urinary pH levels and determining direct tissue damage to the bladder mucosa [30]. Consequently, the precipitation of polyvalent ions (Mg+, Ca+) can occur, with the crystallization of compounds such as magnesium ammonium phosphate and carbonate apatite. The accumulation of crystalline matrix on bacterial biofilm can give resistance to external agents. The presence of crystalline biofilm on catheters (encrustation) can also increase the risk of obstruction and outflow blockage (Figure 1). Additionally, crystalline precipitates can enlarge and aggregate to create urinary sludge and, ultimately, bladder lithiasis [31].

Among the possible complications associated with indwelling catheters, antibiotic resistance surely deserves a mention. The bacteria most frequently related to antibiotic resistance were *Enterococcus faecium*, which are vancomycin-resistant in 85% of cases, and *Pseudomonas aeruginosa*, which are resistant to cephalosporins and aminoglycosides in more than 25% of cases, to fluoroquinolones in more than 34% of cases, and to carbapenems in more than 24% of cases [14]. An animal study compared the efficacy of ciprofloxacin in the treatment of UTIs caused by strains of *P. mirabilis* and *P. aeruginosa*, either in catheterized or non-catheterized mice. Planktonic bacteria inoculated in the bladder of a non-catheterized group was rapidly eradicated by antibiotics. Conversely, in catheterized animals, ciprofloxacin was less effective, resulting in persistent and recurrent colonization [32].

## 6. Non-Antibiotic Prophylaxis and Other Preventive Methods

Little attention is devoted to reducing unnecessary catheterization, and bodies of evidence supporting educational interventions are still too heterogeneous and derived from incoherent results [33].

The bundle concept was developed by the Institute for Healthcare Improvement (IHI) as support for healthcare professionals to improve the care of specific high-risk patients. The bundle is a set of evidence-based interventions that, when properly applied, may greatly improve the outcome of treatment, compared with the use of a single practice. However, applying bundles does not exclude the possibility of adopting additional evidence-based practices that might help prevent CAUTI. A care bundle to prevent CAUTI mainly consists of multiple interventions to improve clinical indications—namely, choosing the right medical aids and equipment, ensuring hygiene and proper drainage, identifying a timeline for catheter removal, or whether any alternatives may be offered in patients suffering from chronic urinary retention and/or untreatable urinary incontinence [34,35]. An updated Cochrane has recently highlighted the need for a standardized set of core outcomes, which should be measured and reported by future studies comparing different approaches for the removal of short-term indwelling urethral catheters in order to reduce the risk of CAUTI and the need for recatheterization [36].

Considering phytotherapy, although evidence supports its use in uncomplicated urinary tract infections, its role in CAUTI prevention is still controversial. Two randomized placebo-controlled trials investigated the efficacy of cranberry in women receiving perioperative urinary catheters. Gunnarsson et al. found no statistically significant difference in the rate of bacteriuria among patients taking 550 mg of cranberry twice a day vs. control [37]. Conversely, Foxman et al. showed a lower occurrence of clinically diagnosed UTIs, with or without positive urine culture, in the interventional group compared with the placebo group [38]. There are few studies investigating cranberry efficacy in long-term indwelling catheters [39]. The largest randomized study comparing the outcomes of cranberry versus methenamine hippurate in spinal cord injury patients has not demonstrated any significant differences in the occurrence and relapse of symptomatic UTI when compared with placebo [40].

Interestingly, a recent observational study reported the successful outcomes of a fixed herbal combination containing *Tropaeoli majoris* and *Armoraciae rusticanae radix*. According to the authors, its use, alone or added to antibiotics, showed a significant reduction in CAUTIs (50% in each group), in contrast to the purely antibiotic group (79%) [41].

Phè et al. investigated the effectiveness of oral D-mannose in neurological patients using or not intermittent catheters (ICs) [42]. The number of monthly proven UTIs decreased both in catheter users and non-users (*p* < 0.01) at 16 weeks of follow-up. At the end of the study, the compliance rates for using D-mannose and dipsticks for testing suspected UTIs were 90.2% and 100%, respectively, in those managed or not by IC.

## 7. Bladder Irrigation and Endovesical Instillation: History, Current Status, and Future Perspectives

Irrigation or washouts of the bladder with various types of bactericidal preparations including antibiotics were introduced in the early 1960s to prevent CAUTI [43,44]. When closed urinary drainage was adopted, its use was significantly decreased and less supported, due to the poor quality of evidence and possible harms reported by some authors. In 1980, Gelman et al. found that those patients receiving irrigation either once a week or twice a day had a higher rate of catheter-associated septic episodes, compared with the group that received no irrigation [45]. Moreover, Elliot et al. suggested potential further damage to the urothelium induced by bladder washouts since an increase in the exfoliation and shedding of urothelial cells was observed, particularly with chlorhexidine or noxythiolin [46]. In addition, bacterial resistance to chlorhexidine was described [47]. By contrast, several studies seem to support the use of polyhexanide, which may combine a broad antimicrobial spectrum with a low risk of resistance, high-tissue compatibility, and good tolerability also in neurological patients with severe spasticity or a history of autonomic dysreflexia [48,49].

A Cochrane review updated in 2017 evaluating the choice, acceptability, complications, and efficacy of different washout regimens concluded that evidence was inconclusive to result in any recommendations, confirming the need for rigorous high-quality trials to clarify which type of solutions, volume, and washout frequency could be helpful in managing long-term indwelling catheterization in older adults [50].

After those findings, several well-designed studies have attempted to overcome the lack of evidence. A recent randomized controlled trial (RCT) including 60 comatose patients has documented a significant decrease in urine colony-forming units, body temperature, erythrocyte sedimentation, and white-cell counts in patients who received bladder irrigation using 450 ccs of normal saline once a day for 3 consecutive days, compared with the control group treated with routine catheter care, supporting short-term efficacy in preventing CAUTI in hospitalized frail patients [51].

A potential way forward is the use of intravesical antibiotics, which have shown to have a greater effect on bacteria at a local level with fewer adverse events [52]. According to a systematic review, gentamicin at various dosages has been the most common antibiotic used intravesically as prophylaxis or treatment, also for neurological patients managed by intermittent or indwelling catheters, with efficacy ranging from 66% to 100% [53]. Similar success was also observed in patients treated with neomycin/polymyxin or colistin but with a higher rate of discontinuation, compared with the gentamicin group. Abrams et al. have proposed a protocol for neurological adults starting with 80 mg gentamicin dissolved in 50 mL of sterile water or 0.9% sodium chloride and left overnight until the patient felt the need to empty, depending on individual bladder capacity [54]. The length of time using gentamicin ranged from 2 to 67 months, and no patient stopped treatment because of raised gentamicin serum levels. Preliminary studies reported good feasibility and success due to its safety regarding the low risk of systemic absorption, nephrotoxicity, or ototoxicity, compared with systemic administration [55]. However, the risk of microbiome alterations induced by chronic antibiotic instillations remains an important issue to clarify [56].

Methenamine is a urinary antiseptic that is hydrolyzed to formaldehyde in acid urine, which should not negatively influence the patient’s intestinal flora and the development of bacterial resistance. This solution was introduced in 1895 for the treatment of UTIs but is today mainly used prophylactically. A significantly low incidence of symptomatic UTI (2.7%) was seen among women receiving methenamine hippurate as prophylaxis after gynecological surgery, compared with placebo [57].

The concept of microbiome manipulation to promote health was introduced several years ago, but only recently has the use of intravesical probiotics been supported by evidence. A phase I study recently published has shown that intravesical *Lactobacillus rhamnosus GG* is a safe route of administration for children with neurogenic bladder as well [58]. The ability of probiotics to adhere to host cell-binding sites is one of the main benefits of probiotics. However, by in vitro studies comparing the inhibitory activity of single probiotics vs. strain mixtures toward pathogenic bacteria, authors concluded that no significant difference has been seen in the ability of single- and multi-strain probiotics to inhibit biofilm formation or reduce the number of cellular adhesions. Therefore, it is still not clear whether an additive or synergistic effect exists between mixed strains, and if it may lead to better outcomes [59].

Another potentially protective intravesical approach is the anti-infective efficacy of bacterial interference of nonpathogenic strains of bacteria such as *E. coli*, which may competitively adhere to the bladder wall and modulate the immunological response. Nonetheless, the identification and development of avirulent strains that effectively and safely outcompete uropathogens is still a challenge in long-term indwelling catheters [60,61].

Despite the introduction of bacteriophages to treat bacterial infections several years ago, their use was consideringly abandoned during the antibiotic era [60]. Only recently has an interest in their efficacy been greatly renewed. Compared with antibiotics, they are highly specific. This is considered an advantage in preserving the microbiome, although it means that an individualized cocktail of bacteriophages is ideally needed to prevent the most common causes of CAUTI [60].

## 8. Conclusions

Determining how and when to remove a catheter and disrupt its lifecycle represents the main challenge for elderly and fragile patients in long-term care institutions.

Although many strategies have been proposed over the decades to prevent symptomatic CAUTI, there is still an urgent need for clinical evidence. Looking back at the author’s “*common sense*” treatment proposals from the past to the present may help us to clarify and understand which “*rational*” options deserve more attention for future research.

## Figures and Tables

**Figure 1 jcm-11-03415-f001:**
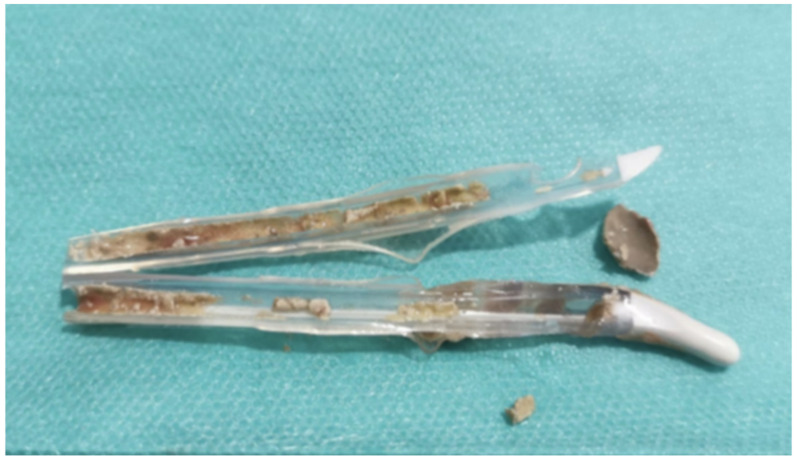
Catheter encrustations in 82-year-old male patient unfit for unobstructive prostatic surgery and chronically managed via indwelling silicone catheter (Tiemann tip).

**Figure 2 jcm-11-03415-f002:**
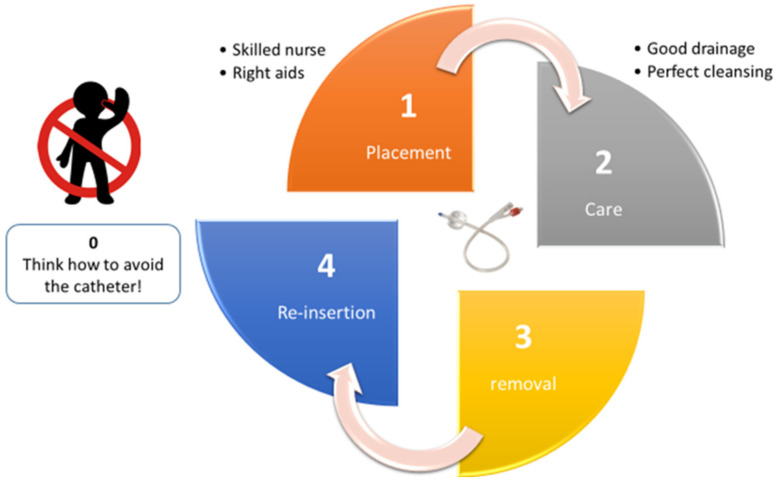
**Catheter’s lifecycle**: (1) consider when and why catheter placement is necessary; (2) daily management and care when catheter is inserted; (3) before catheter reinsertion, consider whether it may be definitively removed; (4) reduce the need for long-term indwelling catheter management.

## Data Availability

Not applicable.

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
