# Peer review of "How to Prevent Catheter-Associated Urinary Tract Infections: A Reappraisal of Vico’s Theory—Is History Repeating Itself?"

_jcm, 2022, doi:10.3390/jcm11123415_

Round 1

Reviewer 1 Report

Minor corrections in the manuscript text must be performed to increase its quality.

line 34 is:... catheter associated.., should be:.. catheter-associated..; check throughout the text

line 42 is:... for the management urinary incontinence.., should be:.. for the management of urinary incontinence..

line 45 is:.. instillation.., should be:.. installation..

lines 48, 65, 108, 110 is:.. UTI.., should be:.. UTIs..

line 57 is:.. CAUTI account.., should be:.. CAUTI accounts..

line 69 is:.. urineculture.., should be:.. urine culture..

line 69 is:.. 105.., should be:.. 105..

line 83 is:.. are present.., should be:.. is present..

line 85 is:.. diagnosis.., should be:..the diagnosis..

line 83 is:.. population.., should be:.. populations..

line 102 is:.. Gram – bacteria.., should be:.. Gram-negative bacteria..

line 103 is:.. Escherichia Coli.., should be:.. Escherichia coli..

line 106 is:.. K. Pneumoniae.., should be:.. Klebsiella pneumoniae..

line 106 is:.. invasion to urothelium.., should be:.. invasion to the urothelium..

line 108 is:.. Gram+ bacteria.., should be:.. Gram-positive bacteria..

line 109 is:..S. Aureus.., should be:.. Staphylococcus aureus..

line 110 is:..S. Aureus.., should be:.. S. aureus..

line 113 is:.. for the bacterial.., should be:.. for bacterial..

line 117 is:.. patient.., should be:.. the patient..

line 128 is:.. with terminal.., should be:.. with a terminal..

line 142 is:.. Ideally a catheter.., should be:.. Ideally, a catheter..

line 152 is:.. According to Meddings et al.. - provide the citation number

line 156 is:.. are: 1).., should be:.. are 1)..

line 160 is:.. attention about.., should be:.. attention to..

line 182 is:.. P. Mirabilis.., should be:.. Proteus mirabilis..

line 183 is:.. Gram- bacterium.., should be:.. Gram-negative bacterium..

line 195 is:.. resistances.., should be:.. resistance..

line 195 is:.. Enterococcus Faecium.., should be:.. Enterococcus faecium..

line 196 is:.. Pseudomonas Aeruginosa.., should be:.. Pseudomonas aeruginosa..

line 198 is:.. some studies.. - Authors provide only one literature item -31. Complete with more citations.

line 201 is:.. of P. Mirabilis and P. Aeruginosa.., should be:.. of P. mirabilis and P. aeruginosa..

lines 224, 225 is:..their.., should be:..its..

line 233 is:.. Tropaeoli majoris and Armoraciae rusticanae radix.., should be:.. Tropaeoli majoris and Armoraciae rusticanae Radix..

line 238 is:.. 16 weeks follow-up.., should be:.. 16 weeks of follow-up..

line 263 is:..RCT.. - explain this abbreviation

line 270 is:.. have greater.., should be:.. have a greater..

line 273 is:.. catheter.., should be:.. catheters..

line 274 is:.. treated by.., should be:.. treated with..

line 287 is:.. L. Rhamnosus GG.., should be:.. Lactobacillus rhamnosus GG..

References - They should be prepared according to the instructions for the authors. https://www.mdpi.com/journal/jcm/instructions

Author Response

Dear Reviewer, thanks for your punctual and deep peer review.

We followed all your instructions and we edited references according your suggestion. Regarding line 198 we preferred to delete the sentences because redundant, rather than adding references as you requested

Please, see the new version of the manuscript.

Reviewer 2 Report

In this review, the authors discuss the current evidence regarding catheter-associated urinary tract infection prevention.  CAUTIs are a significant healthcare burden, and their diagnosis and management remains challenging.  I have several suggestions to improve the manuscript:

1. Line 62 reads "Most cases of nosocomial CAUTI are asymptomatic bacteriuria".  The wording here should be adjusted as asymptomatic bacteriuria is not considered an infection (as the authors indeed note on line 68). Consider "most cases of bacteriuria in the context of an indwelling catheter are asymptomatic".

2. Line 69 - reformat 105 to 10^5 CFU/ml

3. Line 69 - "urine culture"

3.  There are a several areas wherein a reference to support a given statement is not included with that statement.  These include line 94-95, line 113-114, line 133-134, line 179-180, 272-273.  In some of these cases, the citation follows 1-3 sentences later, but this should be adjusted to follow the first statement referencing the study.

4. Authors should italicize bacterial species throughout e.g. line 110, line 195

5. Authors should change "Gram - " to "Gram-negative"

6. Line 112-113 the authors state that microbes can adhere to internal and external aspects of urinary catheter, but do not provide citation.  They should reference the recent study wherein biofilms were stained using crystal violet from catheters of varying indwelling times, showing adherence to all surfaces (PMID: 31430245).  

7. The authors should mention bacterial interference strategy for CAUTI prophylaxis and discuss the relevant literature (26048203, 21683991, 35402319) 

8.  The authors should consider discussing bacteriophage therapy and its potential use going forward in CAUTI prevention and management, as well as the relevant literature and reviews: 32949500, 31732462, 35402319)

9. The catheter lifecycle must be described in text.  Citation should be provided after line 152-153 for Meddings et al study.  Then, each stage should be described within the text itself.  This is particularly important, as they later (line 168) reference "Stage 3", but do not provide any framework for such stages prior to this.  They should also provide a descriptive figure legend for figure 2, providing information regarding each stage.

10.  Can the authors comment on the current status of methenamine irrigation and its evidence? PMID 12174159

11.  The authors should also include reference of the recent review of cranberry juice efficacy in CAUTI: Ji et al Current Bladder Dysfunction Reports volume 15, pages 303–307 (2020).

12. Please provide a more descriptive figure legend for figure 1.  Include patient's age, gender, indwelling time, catheter material, and composition of encrustation (if any/all of this information is available).

Author Response

Dear Reviewer, 

thank you so much for suggesting to us important articles to include in our review which absolutely take into consideration several new aspects of treatment target. 

We reply you in detail according your list.

1) we edited the sentence as you stated

2)done

3)done

4) all the manuscript has been revised and the references are also in brackets

5)done

6)we added reference required

7) and 8) these topics now have been included in the chapter 7 highlighted in yellow and we added 2 references among those that you suggested 

9)done, please see the legend below the figure 2

10)done in chapter 7

11)done (reference n. 39)

12)done, all the available data has been inserted.

We hope that you will be happy with our editing.

Best regard 

Reviewer 3 Report

Thank you for the opportunity to review this manuscript.

The title is very catchy and appealing. However, there is not a lot of new content and there is very little association with the Vico’s theory. Some of the references are very outdated. Moreover, some of the management/prevention options are not discussed in your manuscript (e. g. estrogens, nanoparticles, vaccines, vitamin C). A lot of similar reviews have already been published. 

Some minor comments and suggestions:

  • Some grammar editing is needed (e. g. line 69 – urine culture, line 61 – USD instead of US$, line 103 – Escherichia coli instead of Escherichia Coli, line 109 – aureus instead of S. Aureus etc.).
  • References are not cited properly in the text – the superscripted number is usually just after punctuation mark and not before it.
  • There is some inconsistent use of abbreviations.
  • Line 48-52: Please specify which country you are referring to.

Author Response

Dear Reviewer,

thanks for your comments.

The choice to include outdated references crossing with new current treatment  strategies was "a priori" designed. 

We absolutely agree that this topic is not original but there is still a large interest among clinicians to better understand differences and evidences between several and varied strategies to control CAUTI. Thus we believe our contribute could be helpful for geriatrics, rehab doctors and urologists involved in long-term care of patients suffering from urinary incontinence and/or retention.

line 69: done

line 61: done

line 103: done

We edited the references according to yours and other reviewers suggestion

Kindest Regard